# MaD GUI: An Open-Source Python Package for Annotation and Analysis of Time-Series Data

**DOI:** 10.3390/s22155849

**Published:** 2022-08-05

**Authors:** Malte Ollenschläger, Arne Küderle, Wolfgang Mehringer, Ann-Kristin Seifer, Jürgen Winkler, Heiko Gaßner, Felix Kluge, Bjoern M. Eskofier

**Affiliations:** 1Machine Learning and Data Analytics Lab, Department of Artificial Intelligence in Biomedical Engineering (AIBE), Friedrich-Alexander-Universität Erlangen-Nürnberg (FAU), 91052 Erlangen, Germany; 2Department of Molecular Neurology, University Hospital Erlangen, Friedrich-Alexander-Universität Erlangen-Nürnberg (FAU), 91054 Erlangen, Germany; 3Fraunhofer IIS, Fraunhofer Institute for Integrated Circuits IIS, 91058 Erlangen, Germany

**Keywords:** python, time series analysis, annotation, graphical user interface, gait analysis

## Abstract

Developing machine learning algorithms for time-series data often requires manual annotation of the data. To do so, graphical user interfaces (GUIs) are an important component. Existing Python packages for annotation and analysis of time-series data have been developed without addressing adaptability, usability, and user experience. Therefore, we developed a generic open-source Python package focusing on adaptability, usability, and user experience. The developed package, Machine Learning and Data Analytics (MaD) GUI, enables developers to rapidly create a GUI for their specific use case. Furthermore, MaD GUI enables domain experts without programming knowledge to annotate time-series data and apply algorithms to it. We conducted a small-scale study with participants from three international universities to test the adaptability of MaD GUI by developers and to test the user interface by clinicians as representatives of domain experts. MaD GUI saves up to 75% of time in contrast to using a state-of-the-art package. In line with this, subjective ratings regarding usability and user experience show that MaD GUI is preferred over a state-of-the-art package by developers and clinicians. MaD GUI reduces the effort of developers in creating GUIs for time-series analysis and offers similar usability and user experience for clinicians as a state-of-the-art package.

## 1. Introduction

Machine learning algorithms are a promising approach for different areas in medical research and clinical practice [1,2,3]. They can support clinical workflows or decision-making in diagnosis, treatment, rehabilitation, or prognosis [2,4,5,6]. One part of the development of such algorithms is training or learning. In the case of supervised or semi-supervised machine learning, this necessitates annotated data, which is a time-consuming process [2,7]. For annotating data, the developers of the algorithm or domain experts assign a ground truth value to individual data points. In the domain of gait analysis, this could refer to annotating walking bouts as being part of a standardized gait test [8] or whether a gait pattern is physiological or pathological [9]. The annotated data can then be used to train the algorithm. Furthermore, the annotations can be used to evaluate the algorithm’s performance regarding unseen data.

The segmentation and annotation of time-series can be performed using graphical user interfaces (GUIs) [10,11,12,13,14,15,16,17,18,19]. Although some existing GUIs process similar data with similar goals, for example the annotation of human activities from videos and wearable sensors, they have been developed independently of each other [15,16,17,18,19]. This means the existing code is not re-used within the community, and thus, similar functionalities are implemented in numerous ways, leading to duplicated work. Furthermore, this increases the likelihood of incompatibilities resulting in a barrier to the adoption of technologies [1,20].

One of the reasons for code not being re-used includes code not being publicly available [15,16,17,18]. According to a study conducted by Stodden et al. [21], the major reason for not publishing code is that researchers declare their code as not being cleaned up and undocumented. Another reason for not re-using the mentioned GUIs is that many of them are not written in Python, although Python is the dominant language in scientific computing, especially for machine learning [22,23]. Thus, it can be expected that developers of GUIs for time-series analysis would prefer Python as a programming language and therefore discard available GUIs that have not been written in Python. Furthermore, most of the GUIs seem not to be adaptable to other data formats and/or algorithms and, thus, can only be used with the data and algorithms from the original publication [12,13,14,15,16,17,18,19].

Although some authors mention usability or efficiency to be important [10,11,17], to the best of the authors’ knowledge, for only one of the GUIs from [10,11,12,13,14,15,16,17,18,19], a usability study was conducted. The *Wearables Development Toolkit* (WDK) by Haladjian [11] was tested by three students without prior knowledge of the programming language. Qualitative feedback was collected using semi-structured interviews. The feedback was then used to improve the WDK. Afterwards, a user study with two participants was conducted. For evaluation purposes, an unstructured interview was conducted and seven questions were answered using a five-point Likert scale. However, the sample size of two participants is very small. For a different GUI developed by Fedjajevs et al. [10] named *Platform for Analysis and Labeling of Medical time-series* (PALMS), no explicit results regarding usability were reported. However, the GUI includes algorithms to lower the necessary effort for labeling. These automatically detect characteristic points in an electrocardiogram (ECG) or a photoplethysmogram (PPG) and, thus, decrease the necessary effort for labeling. Likely, this increases usability; however, usability was not evaluated in the study. Furthermore, the authors state that the GUI is adaptable to load data of different formats and use different algorithms. However, evaluations regarding such adaptations are not described in the publication.

Missing evaluations can lead to technical roadblocks, since systems that are difficult to use for domain experts, such as healthcare workers, discourage them from using them [24]. Ultimately, this hinders the translation of developed systems to domain experts or end-users, for example in the domains of sports and health [11,25,26]. More detailed information about technical roadblocks for clinical practitioners was collected in a study by Routhier et al. [26]. They conducted a study on clinicians’ perspectives on factors hindering the data analysis and visualization of time-series data recorded with a wearable sensor. The clinicians most frequently stated that high ease-of-use, as well as the quick generation of results were important to them, underlining the need for testing usability and user experience.

However, usability and user experience have not been tested in existing packages for time-series analysis. More importantly, only one of the GUIs (*PALMS* [10]) listed above is a Python package, which is adaptable in terms of loading different data formats and customization of labels and algorithms. Nonetheless, we observed shortcomings in terms of the adaptability of PALMS and using it in a domain other than ECG/PPG, such as human activity or gait. Additionally, existing GUIs cannot be used for macro annotations, e.g., walking bouts, and micro annotations, e.g., strides, at the same time, which is a major requirement for adoption in the analysis of human activity or gait.

Therefore, we developed a new Python package, the Machine Learning and Data Analytics (MaD) GUI (see Appendix A for code, documentation, and videos). The main contribution of MaD GUI are: (1) It is open-source and, thus, can in the future be used and adapted by other developers for different domains or clinical use cases. (2) MaD GUI’s plugin architecture allows loading arbitrary data formats and the integration of algorithms. (3) The extensive documentation supports the development of plugins and using the GUI.

As a supplement to publishing MaD GUI, we conducted a small-scale study to assess MaD GUI from the perspective of developers, as well as domain experts represented by clinical researchers. Although related publications did not provide such insights [10,12,13,14,15,16,17,18], we think that this is an important step in providing software that can be re-used in the community.

## 2. Materials and Methods

### 2.1. Requirements

Fulfilling certain requirements is crucial to enable developers and domain experts to use the package [27]. For both, the package must be compatible with major operating systems, such as Windows, macOS, or Unix-based systems. We achieved this by including developers using different operating systems in the development of the package. Additionally, we automatically build standalone versions for Windows, macOS, and Ubuntu upon new releases of the package.

#### 2.1.1. Source Code

**Programming language and packages:** As described in the Introduction, Python has become the dominant language in scientific computing, especially in the area of machine learning [22,23]. Therefore, we chose Python as the programming language, such that most researchers in the domain can make use of the package. Furthermore, the plots in the GUI are based on PyQtGraph since it contains classes for labeling regions or samples, which can serve as base classes for our implementation [28]. Accordingly, we used PySide (https://pypi.org/project/PySide2/ (accessed on 27 December 2021)), the Python bindings for the Qt GUI framework, to create the user interface.

**Readability:** Using consistent formatting is important to the readability of code [29]. Therefore, we decided to use the *black* code formatter [30], which enforces following the *PEP8* guidelines [31]. Furthermore, we used *pylint* [32] for additional guidelines not enforced by *PEP8*, as for example, the number of arguments of a method or function. We note that *black* and *pylint* are opinionated and thus could be replaced by other methods.

**Documentation:** Especially for scientific work, documentation is important to ensure reproducibility [33]. As stated in the Introduction, poor documentation is one likely reason for other GUIs not being publicly available. The documentation is even more important for MaD GUI since members of the community must be able to understand the code, which is a prerequisite for adapting the code or for adding new features [34]. The most important parts of the documentation are how to set up the project, how it works, and guides to performing common tasks [27]. All of these items are part of our documentation [35].

**Barriers to entry:** The developer’s effort to use open-source software is an important aspect [27]. Therefore, future users of the MaD GUI package should only be required to interact with packages and data types they are likely familiar with. We achieved this by constructing the plugins to use mostly Python builtins, such as dictionaries, or well-known Python libraries, such as pandas [22,36]. Additionally, we kept the code complexity of the package low by using a linter. Furthermore, we provide a quick start guide for developers in the documentation, provide exemplary data, and made the package installable via pip.

**Composability:** MaD GUI should provide composability, which increases the possibilities for the adaption and maintainability of the package [37]. We achieved this, on the one hand, by using a minimal set of four dependencies. This way, dependency conflicts are only expected in rare cases when implementing other algorithms. On the other hand, we attained composability on a higher level by developing MaD GUI as a plugin system. This makes it possible to combine plugins, for example a plugin for loading a specific dataset with a plugin algorithm.

**Testing:** To keep all parts of the package working even when it grows, for example by attracting other developers, we maintain an extensive test suite for all the core components of the GUI. It is automatically run on each change and assesses whether new developments break important parts of the code, allowing for quick fixes [38]. As a common metric, the percentage of tested code is evaluated. However, this does not reflect the quality of conducted tests and, thus, needs to be treated with caution.

#### 2.1.2. User Interface

**Design:** A well-designed user interface can shorten the required time to learn the system, decrease the time to complete tasks, and overall, lead to increased user satisfaction [38]. We followed evidence-based guidelines for designing the user interface and hope to encourage other researchers to follow these or other applicable guidelines, especially when sharing software with the community. Some principles to mention are the *Eight Golden Rules* by Shneiderman [38], the *Gestalt Laws of Perception* [39,40], as well as *Fitts’ Law of Human Hand Movement* [41,42]. This is not extensive and gives only an exemplary overview of elements we found not, or not completely, fulfilled by other available GUIs for time-series analysis.

**Functionalities:** Next to the design, the user interface must provide certain functionalities. Here, we focus on the functionalities that are relevant for domain experts from gait analysis. According to Routhier et al. [26], the most important aspect for clinicians in terms of functionality is the speed with which results can be generated. This refers both to the processing time of the computer, as well as the amount of “clicks” needed. In terms of analyzing data from inertial measurement units (IMUs), clinicians need to be able to visualize 24 h datasets. Furthermore, clinicians want to be able to select the desired raw data and apply algorithms to them. When asked for improvements regarding existing technology, they most often named ease-of-use.

Although some of these requirements are met by existing GUIs, we had additional requirements that were missing: In our scenario of gait analysis, it is necessary to label macro activities, e.g., walking or jumping, and micro activities, such as single strides. Therefore, several levels of annotations were necessary. Furthermore, it should be possible to load and save custom data formats. Implementing state-of-the-art algorithms in the GUI should be possible in a short amount of time. In terms of use for domain experts, it should be possible to create a standalone executable. Additionally, when labeling activity data, it may be useful to synchronize recorded sensor data with video.

### 2.2. Implementation of the User Interface

The user interface consists of three areas, as shown in Figure 1: the side-bar on the left, the plot, and the mode buttons at the top of the GUI. They serve different functionalities, which are described on a functional level in the next paragraphs.

The side-bar servers as an interface to plugins, which can be used to load data of a specific format, apply an algorithm to the displayed data, or export the displayed data and annotations. When pressing any of the three upper buttons in the side-bar, the user can select a plugin to use. In the case of the *Load data* button, the user can also select a data file and, optionally, a video to be displayed or already existing annotations to be loaded, as shown in Figure 2.

After loading data, the user can click the *Use algorithm* button to run an algorithm. The MaD GUI package contains two exemplary algorithms: one for detecting stationary moments and one for calculating the energy in the detected stationary moments. For exporting the data, displayed annotations, or calculated features, the user can select a plugin by clicking on the *Export data* button. Additionally, the GUI offers the possibility to *Save displayed data* in a pickle format [43]. This allows loading the displayed data and annotations again later using the *Reload displayed data* button or to load them in other applications.

The plot area is created dynamically depending on the plugin that has been used to load the data. This refers to the number of plots being created and the x-axis labels. For example, in gait analysis, it might be necessary to plot data of several sensors from different body parts, e.g., foot, shank, thigh, and lower back [44]. Each of the sensors would be represented in separate plots. The x-axis label can be set to a datetime representation instead of seconds if the start time of the recording is known.

Buttons at the top of the GUI can be used to change the GUI’s mode. To indicate the active mode, the button changes its color, as can be seen in Figure 1. The green line moves with the mouse arrow and is set by pressing the left mouse button or space. Afterwards, a description of the generated annotation can be set in a dialog window. In the *Edit annotation* mode, the user can change an annotation’s description and boundaries. To remove an annotation, the user has to switch to the *Remove annotation* mode. By clicking the button of the activated mode or by pressing *Esc* on the keyboard, the GUI is in *Investigation mode*, and the user cannot add, change, nor delete annotations mistakenly. In this mode, it is possible to move/zoom in and out the data, execute plugins, and inspect the annotations’ description. When hovering over an annotation, a tooltip with the annotation’s description will be shown. If a video is loaded and displayed in additional to the sensor data, synchronization of video and sensor data can be performed in synchronization mode.

### 2.3. Software Structure

An important goal of this package is composability. In terms of software structure, we achieved this using a plugin architecture, which is explained in the following section. Afterwards, we describe the interface between plugins, data, and plots of the GUI represented as a model–view–controller logic.

#### 2.3.1. Plugins

Developers who use the MaD GUI package can create and inject plugins to extend MaD GUI’s functionality. These plugins can be used to:Import data from a recording device *(Importer)*;Define a label to be used within the GUI *(Label)*;Execute an algorithm, which creates or processes annotations *(Algorithm)*;Export plotted data and/or annotations *(Exporter)*.

Plugins can be combined in development or at runtime, such that it is, for example, possible to use an algorithm with data from different recording devices. Here, we give an insight into how a plugin for an *Importer* can be created. An extensive description of plugins and working examples is a part of MaD GUI’s documentation [35].

For all plugins, developers need to create a class inheriting from one of several base classes provided by MaD GUI. In the case of an Importer, this class is called *BaseImporter*. It has a method to expose its name to the GUI, which is used to represent the Importer in the dropdown menu of the load data dialog; see also Figure 2. Additionally, it has a method *load_sensor_data*, which returns a dictionary containing sensor data as a pandas DataFrame and a sampling rate. The exemplary importer that comes with the GUI is able to read *csv* files:



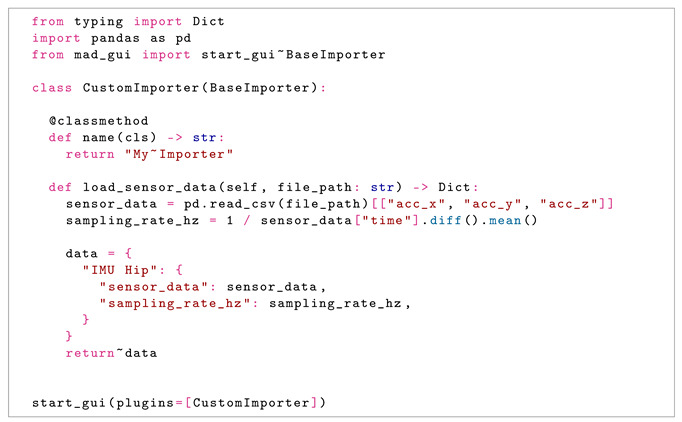



Similarly, as shown in this example, developers can also pass algorithms, configurations for the labels, settings, or a color theme to the GUI. More details on creating these are available in MaD GUI’s documentation [35].

#### 2.3.2. Model–View–Controller

Plugins like an importer or an algorithm use Python dictionaries and pandas DataFrames represent data such that developers of plugins can use familiar data types. However, the underlying libraries for plotting the data need a different representation of the data. Therefore, MaD GUI uses a model–view–controller approach to separate the representation of data from the plotting of the data and from the user interaction [45,46].

**Model:** The model is a representation of all data available in the GUI. Therefore, it serves as an interface between the MaD GUI package and developers of plugins. The model is a *GlobalData* object, which, among other things, keeps the name of the currently loaded file. Additionally, the *GlobalData* object contains one *PlotData* object per plot in the GUI. The *PlotData* object contains the plotted data represented as a pandas DataFrame. Furthermore, plotted annotations are stored in *PlotData’s* attribute *annotations*, represented as a pandas DataFrame. Furthermore, a *PlotData* object can be used to store additional data that are not plotted, but might be necessary for the algorithms to process the data. More information about *GlobalData*, *PlotData*, and *AnnotationData* is available in the package documentation.

**View:** The data and annotations in the *PlotData* objects are plotted by the *view*. Plotting is necessary if data are loaded, an algorithm is executed, or when the user interacts with the view, for example to add or remove an annotation. For plotting, we use *pyqtgrpah*, although matplotlib is the de facto standard for 2D plotting in scientific computing with Python. We chose *pyqtgraph* because it has inbuilt classes for labeling samples or regions, which serve as base classes for the labels that can be plotted. Furthermore, plotting is faster compared to matplotlib [28].

**Controller:** The controller serves as an interface between the user input and the *model*, as well as the *view*. In our approach, there is a global controller, which is the class *MainWindow*. Among others, it handles button clicks and instantiates the views. Furthermore, each plot has one local controller, which changes according to the GUI’s modes. For example, this handles the consequence of a mouse click on an annotation. If the GUI is in the mode *remove*, it gets removed from the view. However, if the GUI is in the mode *investigate*, this would not result in any change.

### 2.4. Code Complexity

The more complex the code is, the harder it is to maintain. Further, a complex software structure makes it harder for new developers to use it. One of the first approaches to assessing the maintainability of software modules was suggested by McCabe in 1976 [47]. He suggested cyclomatic complexity, which is a graph-theoretic complexity measure. It determines the number of independent paths that can be taken through a code and is widely used [48]. The original concept, developed in Fortran, is outdated. This is exemplified by the fact that it does not include structures such as try/catch or lambdas. Furthermore, cyclomatic complexity does not necessarily reflect the effort necessary to comprehend the code [49,50]. Therefore, several models have been developed to assess cognitive complexity, specifically designed to assess code understandability [49,51,52,53]. A well-accepted model was suggested by *SonarSource SA* [54]. Although this model is based on cyclomatic complexity, *SonarSource SA’s* model of cognitive complexity is closer to representing code understandability, which is described in detail elsewhere [55] It is accessible via sonarcloud, but can also be used locally using an extension for *flake8* [56]. For these reasons, we evaluated the packages using this implementation of cognitive complexity.

### 2.5. Study

Similar tools fulfilling parts of the requirements mentioned above for time-series analysis have been published [10,11,15,16,17,18]. However, they were not evaluated regarding adaptability or the user interface for other researchers, except for the *WDK*, which was tested in a pre-study with three participants and a study with two participants. Although studying adaptability and usability is not typical for publishing software packages, we decided to conduct a small-scale study to minimize the potential of severe issues regarding these aspects. The study was conducted among developers to assess the adaptability of MaD GUI from the developers’ perspective and a separate study among clinicians to assess the user interface.

In both parts of the study, we compared MaD GUI to PALMS as representative of the state-of-the-art [10]. We chose PALMS for several reasons: The first reason is that it fulfills our requirement of being programmed in Python. Furthermore, it claims to be adaptable to other data formats or algorithms that are not known to the open-source implementation, which meets another of our requirements. Lastly, the process of annotating data is similar in that both regions and single events can be annotated in MaD GUI and PALMS. In contrast to MaD GUI, PALMS was developed with a focus on ECG and PPG signals and not with a focus on generalization. However, it was the only GUI coming close to our requirements, and thus, we chose to compare MaD GUI with PALMS.

After communication with the local Ethics Committee, this study was exempt from the need to obtain ethical approval. Reasons for this include that we did not expect the study participants to be under higher emotional or psychological stress than in their everyday work. To ensure this, they participated in the study separately and the coordinator of the study did not observe them during the execution of the tasks. Although this study was exempt from ethical approval, participants gave informed consent before they were included in the study.

#### 2.5.1. Adaptability

**Background information:** We included ten participants to assess both MaD GUI and PALMS. The participants had different career stages: four participants were students; four were research assistants or Ph.D. students; two were postdocs. However, since the study was time-demanding, we used a between-group study design. Participants were assigned to either MaD GUI or PALMS. Care was taken to ensure that each group had an equal number of participants from each career stage. Furthermore, we assessed their programming experience and balanced the groups. An approach recommended by Siegmund et al. [57] is to assess self-estimated programming experience compared to peers, as well as programming experience in general. Regarding the latter, all participants had more than two years of programming experience. Specifically for Python, the students had one to two years of experience. The research assistants/Ph.D. students had more than two years of Python experience. One postdoc had 1–2 years of experience in Python and the other one less than one year. The remaining participants had more than two years of experience in Python. Two questions were answered using a five-point Likert scale from 1 (very inexperienced) to 5 (very experienced), of which the results are shown in Table 1:How do you estimate your programming experience compared to your colleagues/fellow students?How experienced are you with object-oriented programming?

The study was conducted remotely at each user’s workplace, on his/her computer. The operating systems were Windows, macOS, and Linux (manjaro). When having certain issues, participants were offered to share their screens and get help from the study coordinator (M.O.). Before conducting the study, several topics were selected for which help was granted: understanding the task, solving dependency conflicts, problems with the development environment, importing classes or functions, or transforming between different data types such as pandas DataFrames and NumPy arrays.

**Tasks:** Before executing the tasks described below, the study coordinator assisted in setting up a virtual environment and the IDE the programmer used. We decided to exclude this from the study since it is the same process for both packages. Afterwards, study participants had to solve four tasks.

First, they were given a link to the GitHub page of the https://github.com/mad-lab-fau/mad-gui#readme (accessed 26 October 2021), https://github.com/PALMS-gui/PALMS#readme (accessed 26 October 2021). The objective was to get an overview of the documentation, such that they get an impression of the package and learn about its documentation structure. Participants could take as much time for this task as they felt necessary, but were informed that they can go back to the documentation any time they felt it was necessary while conducting the remaining tasks.

A low entry barrier is important, as stated in the Introduction. Developers may wish to test the package using example data to get a feeling of whether it suffices for their needs. Thus, installing and starting the GUI is an important aspect. Therefore, this was the second task for developers.

Thirdly, the developers were asked to use the package to load a data format that is not natively supported by the package. The data used in this study were recorded using wearable sensors, which are increasingly adopted in patient monitoring [4]. We used data published by Weiss et al. [58] available via PhysioNet [59]. After the study participants implemented the necessary code, they were asked to show the resulting plot in the GUI to the study supervisor to validate the result.

For the fourth task, developers were asked to implement an open-source algorithm for gait analysis in the GUI. For this task, they were instructed to additionally install PDkit [60]. The developer’s goal was to apply PDkit’s Bellman segmentation to gait data to divide them into two segments. Since we wanted to exclude the time for comprehending PDkit’s interface from this study, we provided them with a function that applies this method to data. Therefore, the users were required to pass data as a pandas DataFrame and the sampling frequency as a float to the mentioned function. It then transforms the data into the PDkit data format, constructs the object for segmentation, and returns results as a human-readable pandas DataFrame. After implementing the necessary code, they were asked to show the resulting plot in the GUI to the study supervisor to validate the result.

#### 2.5.2. User Interface

**Background Information:** For this part of the study, the domain of gait analysis was chosen. However, the only important aspect is to include participants without programming knowledge. The reason is that the user interface should also enable persons without programming knowledge to use open-source, state-of-the-art algorithms. We argue that it is not necessary to have the user interface also evaluated by developers because they likely have a higher technical affinity and will get along easier than users from other domains.

We included six participants in the evaluation of the user interface. Here, we used a within-group study design, since the tasks were less time-demanding compared to the adaptability study. However, half of the group started with MaD GUI, while the other half started with PALMS. Both groups contained a student, a physiotherapist or study nurse, and a medical doctor. All participants had prior experience with recording sensor data. Therefore, they were familiar with IMUs. However, they were given short information on what signals of motion and rest look like. In an attempt to reduce bias, participants were told that both GUIs had been developed by the study coordinator. Upon finishing the study, the participants were informed that PALMS existed before the study and was developed by another group.

Both MaD GUI and PALMS were slightly modified for the user interface study. It was necessary to enable them to load the data of an IMU and to execute the segmentation algorithm from the adaptability study. Furthermore, both packages were adapted such that time spent in the GUI and time spent elsewhere was recorded, for example in the task description or package documentation.

The study was conducted at the university hospital Erlangen, Germany. During the execution of the tasks, the study coordinator did not watch the participants’ actions, unless asked for help. Support was given if participants did not find certain information after searching for more than five minutes and asked for help. Furthermore, questions regarding the necessary accuracy of annotating data were answered.

**Tasks:** Before executing any task, the participants were shown how to start the GUI via the IDE. Furthermore, they received information about how to switch between the browser, the task description, and the GUI. Furthermore, they could ask questions regarding clarification of the tasks.

Before the study, two datasets were recorded by the study coordinator (M.O.). An IMU was worn in the subject’s pocket, who first walked at normal speed, then slow, and then fast. In between, the subject took a short break. In the second recording, the subject walked back and forth a fixed distance four times, referred to as the 4 × 10 m gait test.

The first task was the same as in the adaptability study—to get an overview of the package’s documentation. Participants could spend as much time as they wanted. No questions were answered during this task.

For the second task, the participants had to load data and create manual annotations. In the case of MaD GUI, the task description named the importer they should use, and in the case of PALMS, it provided the database name. They loaded the data of the subject walking at three speeds and were instructed to annotate them accordingly. They could freely switch between the GUI and the task description or GUI documentation if it was necessary. This task was used to assess how easy it was to create annotations in the GUIs.

Lastly, they had to use an algorithm to create automated annotations for the 4 × 10 m gait test. Upon execution of the PDkit algorithm, four segments for walking and four segments for standing were created. The participants had to delete annotations where the subject was not walking. Additionally, they had to adapt the boundaries for the annotations of walking, since most of the automatically created annotations missed one or two gait cycles in the end. This task assesses the usability of editing and removing annotations, but also whether it is easy for users to understand how to edit and remove annotations.

#### 2.5.3. Evaluation

Regarding objective code analysis, we determined cognitive complexity, number of comments, and lines of code. On the one hand, this limits the comparability to projects written in other languages. For example in Java, Weighted Methods per Class, Coupling Between Objects, or Lack of Cohesion in Methods are the most frequently used object-oriented programming metrics [61]. On the other hand, we chose the metrics mentioned above since these are accessible via Python packages, such as radon or prospector. This makes it more likely for our results to be comparable to other Python projects.

All tasks were evaluated using the task completion time. The time needed to fulfill each task was stopped by the supervisor of the study. However, in the case of the adaptability study, we excluded the time needed for installing dependencies. We decided on this since it only led to dependency conflicts with PALMS when additionally installing PDkit. These conflicts can be fixed quickly, by using less strict dependency management in PALMS or PDkit and, therefore, were excluded from the evaluation of the package itself.

We suggest that the task completion time indirectly assesses factors such as good readability, usefulness, structured documentation, and low barrier to entry. However, since it does not yield information about the issues participants encountered during the study, we decided to additionally offer the participants to give unstructured free-text feedback.

After completing all tasks, the study participants filled out two questionnaires. The System Usability Scale (SUS) was used to assess the overall impression of the package [62]. Furthermore, user experience was assessed with the user experience questionnaire [63].

Certain requirements for the MaD GUI package were not assessed directly in the study. Some requirements are fulfilled inherently, for example readability, which is ensured by using a linter and a formatter. Furthermore, a low barrier to entry is partly fulfilled because only Python builtins or well-known Python packages need to be used by the developers. Therefore, these factors were not assessed directly. Instead, we assessed them indirectly using the task completion time. The reason is that code comprehension does not only depend on factors such as objective readability, but is individual, as it depends on factors such as cognitive speed [64]. However, task completion time does not indicate which aspects are causing potential issues, so we additionally obtained unstructured, free-text feedback from the study participants.

## 3. Results

The following section reports results regarding the code of MaD GUI and PALMS. Afterwards, we describe the results of the adaptability and user studies. We are aware that the comparability of MaD GUI and PALMS GUI is limited, as the latter was not specifically built for certain tasks used in this study. In no way should our results be considered a direct criticism of the PALMS GUI, and we encourage authors to consider PALMS as an alternative to our package, especially when using PPG or ECG data.

### 3.1. Code Complexity

We used sonarcloud for code analysis (https://sonarcloud.io/component_measures?id=MalteOlle_mad-gui (accessed 6 February 2022), https://sonarcloud.io/component_measures?id=MalteOlle_PALMS (accessed 6 February 2022)). Selected results are shown in Table 2. MaD GUI roughly has half the cognitive complexity of PALMS. On the one hand, this is attributed to the fact that MaD GUI has 30% fewer lines of code than PALMS. On the other hand, this is influenced by the coding style. Furthermore, the analysis showed that MaD GUI has more comments, both absolutely and relative to the number of lines.

### 3.2. Adaptability Study

This section first describes the time needed to execute the tasks in the study. Afterwards, the results of the assessed questionnaires and free text are presented.

**Time:** Developers using MaD GUI could solve the first task faster than using PALMS, as shown in Figure 3. Regarding the installation and start of MaD GUI, there were no delays on any of the tested operating systems.

Regarding the extension of the packages to load gait data from PhysioNet, MaD GUI outperformed PALMS by twenty minutes, or a factor of two. For PALMS, developers had difficulties understanding the structure of the exemplary database. Furthermore, they had difficulties regarding the data types to be used. Especially, they had to create objects of a specific PALMS in-built class (*Wave*). Before creating this object, they had to understand its structure and working principle. In contrast, for MaD GUI, the developers were required to create a dictionary containing a pandas DataFrame and a float, which are well-known data formats.

The largest difference was found when implementing an open-source algorithm into the GUI. For MaD GUI, the developers needed ten minutes, while for PALMS, they needed 45 min. This large difference resulted mainly from two issues. First, the developers had to find out how to plot annotations (partitions) in PALMS, which was not documented. Second, they had to find out where to execute the algorithm, such that they would have access to the already plotted data. In contrast, for MaD GUI, the developers only needed to fill a method in a documented example and pass the created class to MaD GUI’s function *start_gui*. Similar to loading data, they had to deal with well-known libraries, such as pandas. As an additional contrast to PALMS, plotting was then handled by the MaD GUI package, such that the developers did not need to implement plotting themselves.

For tasks of loading data and implementing an algorithm, we did not include the time needed to install dependencies. The reason is that installing both PALMS and PDkit resulted in dependency conflicts that had to be solved. When installing MaD GUI and PDkit, no dependency conflicts occurred.

**Questionnaires.** The median score for the SUS was 90 and 35 for MaD GUI and PALMS, respectively. The value for MaD GUI is in the third quartile, and the value for PALMS is in the first quartile, referring to quartiles as suggested by Bangor et al. [65]. This suggests better usability of MaD GUI than PALMS from the perspective of a developer extending the package.

Regarding the user experience questionnaire, results are shown in Table 3. MaD GUI received better scores throughout all scales of the UEQ. The lowest rating for MaD GUI was *novelty*, indicating that the design of the overall package could be improved. For PALMS, the lowest rating was obtained for *perspicuity*, indicating that it is hard for developers to get familiar with the system. This is also the score with the largest difference between both packages.

**Feedback:** In free text, the participants stated that both Python packages are useful in general. For PALMS, the developers liked that the readme file is short. Although MaD GUI has a much larger readme file, the developers did not comment on this. Some participants felt MaD GUI’s documentation is well structured and has useful cross-references and helpful working examples, others found the documentation partly confusing. It was noted that there were spelling errors in MaD GUI’s documentation. For PALMS, developers also noted that the documentation was partly confusing. Furthermore, they stated that it could be improved in terms of method docstrings and information about partitions or annotations in general. Several users had issues installing an open-source algorithm and PALMS in the same Python environment due to the strict dependency management of both packages. Additionally, developers disliked that they received error messages, which were not meaningful to them.

### 3.3. User Interface Study

This section follows the same outline as the previous section. First, the time needed to fulfill the tasks is described. Then, we show the results of the questionnaires. The last part describes the written feedback from the participants.

**Time:** The median time to load data and annotate them was below two minutes for MaD GUI and below five minutes for PALMS, as shown in Figure 4. Most participants needed longer for PALMS since they had difficulties finding the documentation for generating annotations in the provided videos. For the second task, using an algorithm for annotations, one participant stated that she did not find the necessary information for PALMS after seven minutes. The participant was then given the hint to watch the video. However, once participants knew how to interact with the annotations, they were nearly as fast as using MaD GUI.

**Questionnaires:** The median score for the SUS was 85.0 for MaD GUI and 62.5 for PALMS. With these results, MaD GUI is in the second and PALMS in the first quartile according to an empirical study by Bangor et al. [65]. This suggests higher usability of MaD GUI’s interface than PALMS interface. The difference between both packages’ scores is much smaller than in the adaptability study. However, PALMS is below the average score for GUIs of 75 [65].

Results from the user experience questionnaire are shown in Table 4. For both packages, the lowest value was *novelty*, indicating that the design of the user interface does not catch users’ interest to a high degree. A large difference of more than two points is evident regarding *perspicuity*. Consistent with the task completion time, this suggests that it is easier to get familiar with MaD GUI than with the PALMS user interface.

**Feedback:** In free-text answers, the users stated that both GUIs offer a good overview of the data. For MaD GUI, it was also noted that it was easy to learn. As a drawback of MaD GUI, the participants mentioned that algorithms are not executed automatically upon loading data, but have to be selected manually by the user. Furthermore, one participant did not like that possible shortcuts were not shown within the GUI. For PALMS, the same user noted positively that shortcuts are shown in the GUI. However, others would rather have liked an overview of shortcuts in the form of a table as it is part of MaD GUI’s documentation. Another participant stated that the shortcuts of PALMS are not intuitive, while the mode buttons of MaD GUI are intuitively usable. For both packages, the participants stated that the documentation was confusing. Opinions about color scheme and contrast differed: some users preferred PALMS and others MaD GUI. In summary, controversial feedback was given with personal preferences for one version or another.

## 4. Discussion and Limitations

In this small-scale study, we compared MaD GUI to PALMS as representative of the state-of-the-art. We used both objective and subjective measures and found the novel MaD GUI to perform superior in the given context. In no way should our discussion be considered a direct criticism of the PALMS GUI, and we encourage authors to consider PALMS as an alternative to our package, especially when using PPG or ECG data.

This study comprises a small sample size. However, according to Virzi [66], as few as five participants are sufficient to find 80% of issues. Therefore, we suggest that our study is useful to rule out the fundamental shortcomings of the MaD GUI package.

Our study showed that the MaD GUI package is easier to use than the state-of-the-art regarding adaptability by developers. MaD GUI has a lower code complexity, indicating that it is easier to maintain and extend. Additionally, the number of comments is higher for MaD GUI than for PALMS. This is backed up by the fact that developers suggested improving PALMS’ method docstrings, whereas no developer suggested this for MaD GUI.

The task completion time was lower for MaD GUI for all tasks in the adaptability study. However, the increased time needed to start PALMS results from the fact that it depends on *pywin32* and, as such, only users of Unix-based operating systems experienced this issue. If this would be fixed, the difference in task completion time would decrease and new developers might need the same time to start MaD GUI and PALMS. However, the more relevant task completion times are those of loading data and implementing an algorithm, since those tasks take the longest to complete. Here, MaD GUI outperforms PALMS. For the task of implementing an algorithm, this is mostly attributed to the quality of documentation, as became apparent from the developers’ feedback. The objective results are further supported by subjective feedback in terms of questionnaires and open feedback. Concluding, the results indicate the superior performance of MaD GUI in this study setting.

Regarding the user interface study, the task completion time for both tasks was smaller for MaD GUI than for PALMS. However, the differences were small, which is, among others, attributed to the short absolute amount of time needed to fulfill the tasks. The results indicate that users need a shorter time to understand the working principle of MaD compared to PALMS. However, for the second task, there was a difference of less than a minute compared to MaD GUI, suggesting that after initially understanding the GUI, domain experts can apply algorithms in an equal amount of time using MaD GUI or PALMS. The subjective feedback in terms of SUS and UEQ also indicates the superior performance of MaD GUI. However, these subjective differences are small considering the free-text answers.

Since our implementation was based on guidelines from the literature and the limitations of existing packages, it is expectable that MaD GUI outperforms the state-of-the-art. However, in contrast to other publications, we are the first to conduct an evaluation of the package in a study, which can serve as a guideline for developers of other tools.

A limitation of this study is that MaD GUI’s documentation was created by the authors of this study. As a result, the authors may have had a special focus on the parts of the documentation that were necessary to conduct the study. This will only become evident if the MaD GUI package is going to be used by more developers.

Furthermore, we did not consider the time developers needed to set up their programming environment, e.g., installing dependencies. We argue that this highly depends on the way project dependencies are handled and on the experience of the developers. Furthermore, changing the way of handling project dependencies can be resolved fairly quickly and, therefore, should not be a major point of concern.

For the user interface study, a reason for the difference in task completion time between MaD GUI and PALMS was that users were not able to find the necessary information in PALMS’ documentation quickly. However, this could be solved in a real-life scenario by the developer explaining the respective GUI or documentation to the potential user. Nonetheless, we decided not to give study participants an oral introduction in order not to bias them in any direction.

Participants of the adaptability study were mostly from the same lab as the authors of MaD GUI. Therefore, they may share similar coding styles, which may have influenced the results in favor of MaD GUI. However, only two of the study participants had been working with the authors of MaD GUI before, and considering, this they were assigned to the PALMS GUI. Additionally, we included two external participants from Universidade do Vale do Rio dos Sinos (Brazil) and Newcastle University (United Kingdom) in the adaptability study to mitigate this issue.

One clinician noted that she had difficulties finding certain information in MaD GUI’s documentation. However, this could be avoided by separating the documentation for the user interface and for developing plugins from each other. One option is to separate the documentation for the user interface from the GitHub main project page, for example to GitHub pages.

Future developments may improve usability and user experience even further, for example by adding information about shortcuts to the user interface. Similarly, displaying tooltips more prominently might reduce the need for users to search for relevant information in MaD GUI’s documentation. In the current implementation, tooltips are shown only when the mouse is not moving for several seconds. The explorative behavior of users continuously moving the mouse might prevent the tooltips from being shown in the current implementation. Although not tested in this study, at the time of writing, issues are known regarding the synchronization of video and sensor data, which does not work for certain types of videos. However, a fix is known and being implemented, as can be seen in the GitHub issue tracking of MaD GUI.

## 5. Conclusions

We created and analyzed a Python package, MaD GUI, for the development of GUIs for time-series analysis. MaD GUI is open-source and, thus, can, in the future, be used and extended by other developers for different domains or clinical use cases. In a study, we showed that MaD GUI’s plugin architecture allows easy adaptation to other data formats and algorithms. Furthermore, MaD GUI is usable for domain experts outside of computer science, e.g., clinical researchers, for annotating time-series data or applying algorithms.

The plugin-based architecture of the package allows it to be used in various contexts and enables easy implementation of (open-source) Python algorithms. In comparison to a recently published package for a similar purpose, MaD GUI enabled both developers, as well as domain experts to fulfill their tasks more rapidly, saving up to 75% of the time. Furthermore, they were more content with the documentation and effort needed to execute the tasks. Our approach shows that considering generic guidelines for the development of user interfaces, as well as the opinions of domain experts drastically increases the package’s quality. On the one hand, we hope that this encourages researchers creating algorithms or GUIs to follow some of these and related principles to increase the reusability and quality of their research. On the other hand, MaD GUI should serve as a package for the community to ease the creation of GUIs for time-series analysis and, therefore, to be able to focus on their actual research topics. The conducted study suggests that MaD GUI performs well concerning adaptability and usability in terms of the tested aspects.

## Figures and Tables

**Figure 1 sensors-22-05849-f001:**
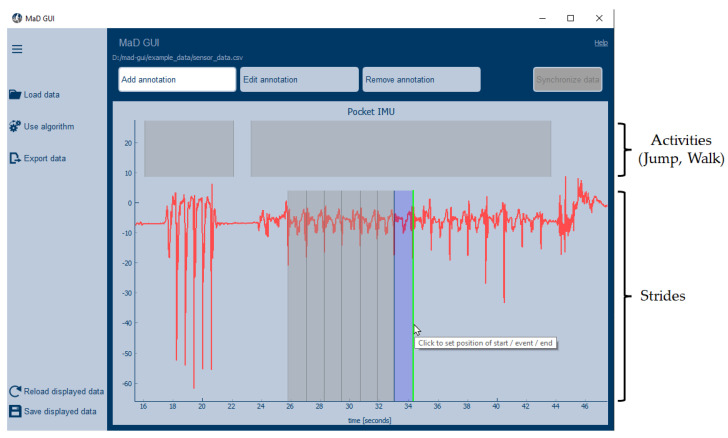
Main window of MaD GUI. The side-bar (left) gives access to plugins, which can be injected into the GUI. The buttons on the top change the GUI’s mode, and the *Add annotation* mode is selected. In the upper part of the plot (first level), activities were annotated. The second level was used to annotate strides.

**Figure 2 sensors-22-05849-f002:**
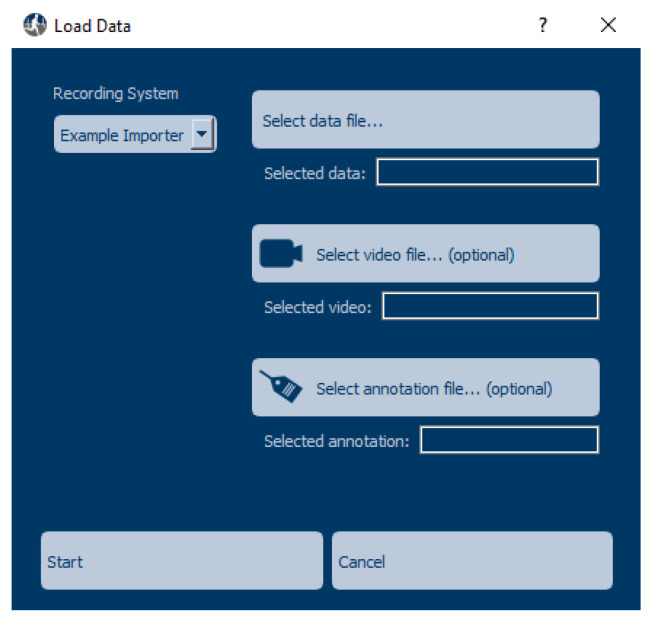
The load data dialog, where the user selects data to be loaded.

**Figure 3 sensors-22-05849-f003:**
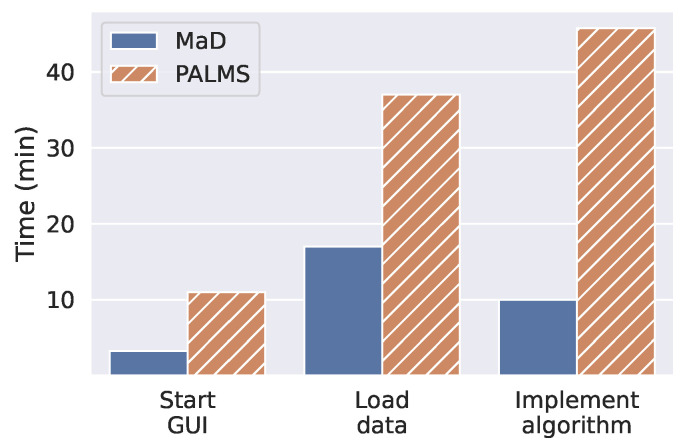
Median task completion time for developers. For the tasks *Load data* and *Implement algorithm*, the time needed for installing the required Python packages is not included.

**Figure 4 sensors-22-05849-f004:**
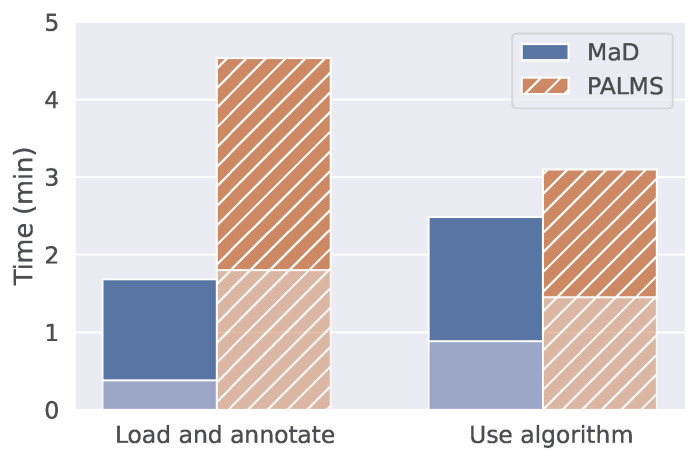
Task completion time for clinicians. Divided into the GUI being in the background (lower opacity) or foreground (higher opacity). Time in the background and foreground are medians over five study participants. For one subject, data recording failed for the task *Load and annotate*; however, according to the manually stopped overall time, this person’s data would not change the median.

**Table 1 sensors-22-05849-t001:** Developer programming experience according to a 5-point Likert scale (1: very inexperienced, 5: very experienced) reported as the mean (median).

Item	PALMS	MaD
Experience with respect to colleagues	3.8 (4.0)	3.4 (3.0)
Object-oriented programming	3.4 (3.0)	3.4 (3.0)

**Table 2 sensors-22-05849-t002:** Static code analysis. Percentage values are relative to lines of code.

Measure	PALMS	MaD GUI
Lines of code	5134	3574
Comments (%)	897 (14.9)	1009 (22.0)
Cognitive complexity (%)	1035 (0.20)	548 (0.15)

**Table 3 sensors-22-05849-t003:** User experience questionnaire median scores for the adaptability study. For each item, a score between −3 (worst) and 3 (best) is possible.

Scale	MaD GUI	PALMS
Attractiveness	2.17	−0.67
Perspicuity	2.25	−2.00
Efficiency	2.50	−0.50
Dependability	2.25	−0.50
Stimulation	2.00	0.25
Novelty	1.75	−1.00

**Table 4 sensors-22-05849-t004:** User experience questionnaire median scores for the user interface study. For each item, a score between −3 (worst) and 3 (best) is possible.

Scale	MaD GUI	PALMS
Attractiveness	1.67	0.33
Perspicuity	2.88	0.75
Efficiency	2.13	1.38
Dependability	1.50	0.63
Stimulation	1.25	0.46
Novelty	0.75	−0.42

## Data Availability

Study data will be made accessible upon reasonable request to the corresponding author.

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
