# Peer review of "MaD GUI: An Open-Source Python Package for Annotation and Analysis of Time-Series Data"

_sensors, 2022, doi:10.3390/s22155849_

Round 1
Reviewer 1 Report
This manuscript reports python for time-series data with annotation analysis. The following issues need to solve before publication.
1, The authors need to cite more papers to support the background about data processing, such as Micromachines. 2022, 13, 847.
2, Is the user interface designed by authors or the comemricia? The authors need to make it clear in the manuscript.
3, What is the difference between Figure 1 and 3? the authors can explain more about the two figures
4, What is difference between PALMS and MaD GUI? Any specific purpose between them?
5, How about any other methods in Table 4?
4,
Reviewer 2 Report
The topic of the article is very interesting, as authors describe an open-source Python package for annotating and analyzing time-series data. There are though some serious issues to consider to improve it:
- Adaptability of the tool (which is stressed in the abstract of the article) is not adequately assessed in the rest of the text and the deployed methodology.
- Factors mentioned in 2.1.1 are not assessed during the developers’ study. Why is that?
- Human-Computer and GUI design guidelines referenced in 2.1.2 are very general. When designing a GUI to ensure usability this is a very strange selection of principles. For instance, you reference Gestalt Laws, and you claim that due to the law of proximity you chose to put certain buttons closely together on the screen. This sounds very simplistic, as this is a rational and rather mainstream approach even for someone that has never heard of this law. I cannot believe that you made this choice only after you studied the law of proximity. Also, you reference the 8 golden rules but again all examples you give are very general and give no practical added-value to your descriptions and your readers. I do not think that there is a point in giving so much length to describe so widely accepted and trivial remarks. The same goes for Fitt’s Law.
- In lines 315-316 something is missing (makes no sense)
- With 10 developers (in a between-subjects setting) and 6 experts (in a within-subjects setting) I would definitely call this study “preliminary”. The number of participants is very low (I am afraid that this could be a reason to reject the article).
- (lines 411, 412) Was the truth restored afterwards? Users have the right to know the truth and this I believe is an ethical obligation of the researchers.
- (lines 420-422) Were these incidents reported and taken into account when you analysed your metrics?
- SUS is not considered reliable for 6 users (you must have at least 12 id I am not mistaken)
- In table 2 you calculate and present 3 metrics. Why did you choose those metrics (lines of code is not considered reliable when comparing different programming styles….)? There are several sets of well-known OO metrics you can use to have a better and more reliable picture of the quality of code.
- In lines 587 and 588 you state that “… none of the available packages fulfilled all requirements described in section 2.1”. This is a very “heavy” claim. How can you prove that other tools have not taken into account -Fitt’s Law or the 8 golden rules, or Gestalt Laws??????
- Lines 593-596 and 608-612: this is a serious limitation as it may greatly bias all your comparative results. It would be better to expand your group of developers otherwise your findings are compromised.
- The article must be proof-read carefully as there are several typos (for instance, line 89 “expertes”, line 239 “importer”)
Round 2
Reviewer 1 Report
The authors did solve all my issues. I recommend to publish at the current form